# On the Feasibility of Fréchet Radiomic Distance–Constrained Adversarial Examples in Medical Imaging: Methods and Trade-offs

**Mohamed Mahmoud**[*1]

MOHAMED.MAHMOUD@RAMYRO.COM

**Shehab Khaled**[*2]

SBEBAB.ELHOUSIENY03@ENG-ST.CU.EDU.EG

**Mohamed Elkhayat**[2]

MOHAMMED.KHAYYAT02@ENG-ST.CU.EDU.EG

**Jamil Fayyad**[3]

JFAYYAD@UVIC.CA

[1] *RAMYRO Inc., Cary, USA*

[2] *Computer Engineering Department, Cairo University, Egypt*

[3] *University of Victoria, Victoria, BC, Canada*

**Editors:** Accepted for publication at MIDL 2026

## Abstract

Adversarial attacks expose critical vulnerabilities in medical imaging AI models; yet, most existing methods violate the textural and structural characteristics that define authentic medical images by disregarding the clinical and radiomic plausibility of the generated perturbations. In this study, we present the first systematic investigation in the *existence and feasibility* of adversarial examples constrained by the Fréchet Radiomic Distance (FRD) a quantitative measure of radiomic similarity capturing textural, structural, and statistical coherence between images. We formulate a gradient-free, multi objective optimization framework based on Multi Objective Particle Swarm Optimization (MOPSO) operating in the Discrete Cosine Transform (DCT) domain. This framework jointly minimizes FRD and maximizes adversarial deviation, allowing a principled exploration of the trade off between radiomic fidelity and adversarial strength without requiring gradient access. Empirical evidence across multiple medical imaging models demonstrates that enforcing strong FRD constraints (FRD $\leq 0.05$) dramatically reduces adversarial feasibility. Perturbations preserving radiomic fidelity consistently fail to achieve meaningful adversarial deviation, suggesting that radiomic realism imposes an intrinsic feasibility boundary on adversarial generation. These findings establish radiomic consistency as a fundamental constraint on adversarial vulnerability, offering theoretical and empirical insight toward the development of inherently robust and trustworthy medical imaging AI. Our code is publicly available here.

## 1. Introduction

Deep learning has achieved remarkable success across medical imaging applications, including disease classification, lesion segmentation, and anomaly detection (Ronneberger et al., 2015; Elkhayat et al., 2025; Henry et al., 2022; Fayyad et al., 2025). However, these systems remain highly vulnerable to *adversarial perturbations*: small, imperceptible noise that can lead to substantial errors in model predictions (Szegedy et al., 2014; Goodfellow et al., 2015). In safety critical contexts such as oncology, neuroimaging, and radiology,

---

[*] Contributed equally

such vulnerabilities raise significant concerns about the *trustworthiness and reliability* of AI models (Finlayson et al., 2019; Maier-Hein et al., 2022; Fayyad et al., 2024).

Conventional adversarial attacks broadly fall into two categories: white-box methods (e.g., Fast Gradient Sign Method (FGSM) (Goodfellow et al., 2015) and Projected Gradient Descent (PGD) (Madry et al., 2017)) which rely on full gradient access, and black-box approaches that utilize gradient-free optimization. These latter methods include directional search techniques like SIMBA (Guo et al., 2019) and evolutionary algorithms like GenAttack (Alzantot et al., 2019). While these attacks are effective, they primarily optimize for pixel space or feature space deviations, critically disregarding the radiomic plausibility of the resulting images. (Ma et al., 2021) showed that adversarial perturbations can systematically alter image statistics and texture cues, leading to perturbed examples that differ from authentic medical imaging characteristics.

In medical imaging, *radiomics* has emerged as a powerful paradigm for quantitatively describing tissue heterogeneity, shape, and texture using handcrafted or learned features. Radiomic features have demonstrated strong clinical associations with diagnosis, prognosis, and treatment response across multiple modalities (Bo et al., 2024). This motivates a fundamental question:

> **Do adversarial examples exist when constrained to preserve radiomic fidelity?**

To investigate this, we employ the *Fréchet Radiomic Distance (FRD)* (Konz et al., 2025) a metric inspired by the Fréchet Inception Distance (FID) (Heusel et al., 2017), but grounded in radiomic feature space to quantify the statistical similarity between original and perturbed images; FRD was chosen over FID because previous work (Konz et al., 2025) has shown that it is more robust and clinically relevant for medical imaging datasets. In particular, FRD demonstrates a higher correlation with radiologist-perceived image quality and downstream clinical task performance compared to FID, and it remains reliable even with the limited sample sizes typical of medical cohorts, whereas FID requires more samples to produce stable scores. A low FRD indicates that the perturbed image maintains radiomic consistency with the original, preserving the appearance of a valid medical image. . A low FRD implies that both images share consistent radiomic representations, maintaining the appearance of a valid medical image.

We introduce a gradient-free *Multi Objective Particle Swarm Optimization (MOPSO)* framework that operates in the *Discrete Cosine Transform (DCT)* domain to explore the trade off between adversarial strength and radiomic fidelity. The optimization jointly:

1. minimizes the FRD to preserve radiomic realism, and

2. maximizes the adversarial deviation in model predictions.

By operating in the DCT domain, perturbations are restricted to perceptually relevant frequency components, ensuring physically meaningful and interpretable modifications (Wang et al., 2020).

Our empirical analysis reveals that under stringent FRD constraints, adversarial feasibility sharply diminishes. Perturbations that maintain radiomic fidelity frequently fail to achieve adversarial deception, suggesting that radiomic realism imposes a natural robustness boundary on model vulnerability.

**Contributions.** This study provides a conceptual and computational analysis of adversarial feasibility under radiomic fidelity constraints. Specifically, we:

- **Formulate the FRD constrained adversarial existence problem.** We define adversarial feasibility as a multi-objective optimization problem balancing radiomic similarity and model deception, establishing a formal link between robustness and radiomic realism.

- **Develop a gradient-free MOPSO framework in the DCT domain.** Our formulation enables systematic exploration of adversarial feasibility landscapes without gradient access, bridging perceptual and statistical perspectives on image realism.

- **Characterize the limits of adversarial feasibility under radiomic fidelity constraints.** Extensive optimization trials reveal that beyond a threshold of radiomic fidelity (low FRD), adversarial objectives collapse, exposing a natural robustness frontier embedded in radiomic space.

## 2. Related Work

Adversarial robustness in medical imaging has become an area of growing concern as deep learning models are increasingly deployed in clinical decision pipelines. Early studies demonstrated that imperceptible pixel level perturbations can drastically alter model predictions (Szegedy et al., 2014; Goodfellow et al., 2015). In the medical domain, such perturbations can lead to critical misdiagnoses (Finlayson et al., 2019; Ma et al., 2021), highlighting the vulnerability of convolutional and transformer-based models used for radiology and pathology. Existing attack formulations typically optimize for minimal perceptual distortion while maximizing model error, employing methods such as FGSM, PGD, or evolutionary optimization for black box setups (Alzantot et al., 2019). However, most of these methods define distortion using pixel space metrics (e.g., $\ell_p$ norms), which fail to align with clinically relevant image similarity.

To address the limitations of pixel based similarity, several works have proposed perceptual metrics and distributional distances to constrain adversarial perturbations. The Fréchet Inception Distance (FID) (Heusel et al., 2017) and its domain specific variants have been adopted as proxies for semantic fidelity in generative models. In medical imaging, recent studies have extended this concept toward radiomic spaces, giving rise to the Fréchet Radiomic Distance (FRD) (Konz et al., 2025). FRD measures the distributional discrepancy in radiomic feature space, reflecting changes in texture, intensity, and morphology that are perceptually meaningful to radiologists. Despite its relevance, FRD has not yet been systematically integrated into adversarial optimization frameworks to assess the feasibility of radiomic preserving attacks.

Incorporating hard or soft constraints into optimization problems has been explored through several paradigms.(Raissi et al., 2019) introduce physics based penalty terms that embed known governing equations directly into the loss, enforcing consistency with physical laws. (Madry et al., 2017) introduced adversarial training, which imposes robustness constraints by optimizing against worst case perturbations, regularizing the model through adversarial examples. Wasserstein based and distribution constrained attacks attempt to bound perturbations under transport based metrics. (Wong et al., 2020). However, enforcing high-level

constraints such as radiomic consistency remains an open challenge, particularly in the black box setting. Multi objective evolutionary algorithms, such as CMA-ES, NSGA-II and MOPSO (Hansen and Ostermeier, 2001; Deb et al., 2002; Coello Coello and Pulido, 2004), provide a natural framework for exploring the trade-off between robustness and constraint satisfaction without explicit gradient information.

While previous research has focused on perceptual or structural constraints, none have explicitly studied whether adversarial examples can exist under clinically meaningful fidelity metrics such as FRD. Our work is the first to formulate and empirically analyze the existence of FRD constrained adversarial perturbations in a gradient-free setting, providing both methodological insights and evidence based conclusions about their feasibility and trade offs.

## 3. Methodology

### 3.1. Problem Formulation

Let $f_\theta : \mathcal{X} \to \mathcal{Y}$ denote a neural network trained for medical image classification or embedding, where $\mathcal{X} \subset \mathbb{R}^{C \times H \times W}$ is the image domain. Given a clean image $x \in \mathcal{X}$, our goal is to find an adversarial perturbation $\delta$ such that the perturbed image $x' = x + \delta$ induces a significant change in the model's feature representation or prediction, while remaining indistinguishable under the Fréchet Radiomic Distance (FRD). This yields the following constrained optimization problem:

$$
\begin{aligned}
\max_{\delta} \quad & \mathcal{L}_{adv}(f_\theta(x), f_\theta(x')) \\
\text{s.t.} \quad & \mathrm{FRD}(x, x') \leq \tau, \\
& \|\delta\|_\infty \leq \epsilon, \\
& x' \in [0, 1]^{C \times H \times W},
\end{aligned}
\tag{1}
$$

where $\mathcal{L}_{adv} = \|f_\theta(x) - f_\theta(x')\|_2$ denotes the adversarial objective (i.e., the L2 distance between embeddings), $\tau$ is a radiomic similarity threshold, and $\epsilon$ bounds the pixel-level perturbation magnitude. The constraint $\mathrm{FRD}(x, x') \leq \tau$ enforces that the generated sample remains within a radiomic-consistent region of the data manifold.

### 3.2. Fréchet Radiomic Distance (FRD)

FRD extends the Fréchet Inception Distance (FID) to radiomic feature space. Let $\phi(\cdot)$ be a radiomic feature extractor computing a feature vector of $K$ statistics (e.g., texture, intensity, shape descriptors) from medical images. Given two sets of images $\mathcal{X}_r$ (reference) and $\mathcal{X}_g$ (generated), the FRD between their feature distributions is defined as:

$$
\mathrm{FRD}(\mathcal{X}_r, \mathcal{X}_g) = \|\mu_r - \mu_g\|_2^2 + \mathrm{Tr}\left(\Sigma_r + \Sigma_g - 2(\Sigma_r \Sigma_g)^{1/2}\right),
\tag{2}
$$

where $(\mu_r, \Sigma_r)$ and $(\mu_g, \Sigma_g)$ denote the empirical means and covariances of radiomic features for $\mathcal{X}_r$ and $\mathcal{X}_g$, respectively. $\mathcal{X}_r = \{x_1, x_2, ..., x_j\}$ and $\mathcal{X}_g = \{x'_1, x'_2, ..., x'_i\}$ where $i$ and $j$ represnet number of samples.

In our implementation, we adapt FRD to operate on single-image embeddings by computing its distance relative to a reference batch of clean samples. Let $\mathcal{X}_r = \{x_1, \ldots, x_N\}$ denote a batch of clean images and let $x_g'$ be a single adversarial image. The corresponding radiomic feature distribution of the adversarial batch collapses to a single point, yielding a zero covariance matrix, i.e., $\Sigma_g = \mathbf{0}$ and $\mu_g = \phi(x_g')$.

Under this setting, the FRD expression simplifies to

$$\text{FRD}(\mathcal{X}_r, x_g') = \|\mu_r - \phi(x_g')\|_2^2 + \text{Tr}(\Sigma_r), \tag{3}$$

where the trace term depends only on the reference distribution and is constant with respect to the adversarial optimization.

Since $\text{Tr}(\Sigma_r) \geq 0$ and is fixed, minimizing the mean deviation term

$$\|\mu_r - \phi(x_g')\|_2^2 \tag{4}$$

provides a lower bound on the full FRD:

$$\|\mu_r - \phi(x_g')\|_2^2 \leq \text{FRD}(\mathcal{X}_r, x_g'). \tag{5}$$

Therefore, optimizing this term directly enforces radiomic consistency with respect to the reference distribution while remaining faithful to the FRD formulation.

## 3.3. Low-Frequency DCT Parameterization

Adversarial perturbations optimized directly in image space often contain high-frequency noise that is imperceptible to humans but can disrupt the radiomic properties of medical images, resulting in unrealistic or clinically implausible patterns. To address this, we parameterize the perturbation $\delta$ in the low-frequency subspace of the DCT, which decomposes an image into a set of orthogonal spatial frequency components.

Restricting perturbations to low-frequency DCT components ensures that they are smooth and structured, avoiding abrupt pixel-level noise that could violate radiomic realism. Additionally, this parameterization reduces the dimensionality of the optimization problem: instead of optimizing over all $C \times H \times W$ pixels, we only search over the $d$ coefficients in $\alpha$, where $d \ll C \times H \times W$.

Formally, the perturbation is expressed as:

$$\delta = B\alpha, \tag{6}$$

where $B \in \mathbb{R}^{(C \times H \times W) \times d}$ is a truncated DCT basis containing the first $d$ low-frequency components of the image, and $\alpha \in \mathbb{R}^d$ is the coefficient vector optimized by our MOPSO framework. Each column of $B$ represents a smooth spatial pattern corresponding to a specific low-frequency DCT mode, and $\alpha$ specifies how strongly each mode contributes to the final perturbation.

## 3.4. Multi-Objective Particle Swarm Optimization (MOPSO)

Since the FRD constraint and the adversarial objective are non-differentiable in the black-box setting, we employ a Multi-Objective Particle Swarm Optimization (MOPSO) strategy

(Coello Coello and Pulido, 2004). Each particle represents a candidate $\alpha$, and the swarm collectively explores the Pareto front balancing radiomic similarity and adversarial strength.

For each particle $i$, its velocity and position are updated as:

$$\begin{aligned}
v_i^{(t+1)} &= wv_i^{(t)} + c_1r_1(p_i - \alpha_i^{(t)}) + c_2r_2(g - \alpha_i^{(t)}), \\
\alpha_i^{(t+1)} &= \alpha_i^{(t)} + v_i^{(t+1)},
\end{aligned} \tag{7}$$

where $w$ is the inertia weight, $c_1$ and $c_2$ are cognitive and social coefficients, and $g$ is a leader selected from the Pareto archive. Each candidate is evaluated via $(\text{FRD}(x, x'), \mathcal{L}_{adv})$, and non-dominated solutions are retained to approximate the Pareto frontier.

### 3.5. Optimization Procedure

The attack process is summarized as follows:

**Algorithm 1:** DCT-based Multi-Objective Adversarial Attack via MOPSO
**Input:** Image $x$, model $f$, particles $N$, iterations $T$, DCT dimension $d$
**Output:** Pareto-optimal adversarial examples
Generate DCT basis $B \in \mathbb{R}^{|\mathcal{X}| \times d}$; initialize particles $\alpha_i$ and velocities $v_i = 0$; set personal bests $p_i = \alpha_i$; archive $\mathcal{A} = \emptyset$;
**for** $t = 1$ **to** $T$ **do**
  **for** $i = 1$ **to** $N$ **do**
   $x_i' \leftarrow \text{clip}(x + B\alpha_i, 0, 1)$; $f_1^{(i)} \leftarrow \text{FRD}(x, x_i')$; $f_2^{(i)} \leftarrow -\mathcal{L}_{adv}(f, x, x_i')$;
  **end**
  **for** $i = 1$ **to** $N$ **do**
   **if** $(f_1^{(i)}, f_2^{(i)})$ *non-dominated w.r.t.* $\mathcal{A}$ **then**
    add $\alpha_i$ to $\mathcal{A}$ and remove dominated entries;
   **end**
  **end**
  **for** $i = 1$ **to** $N$ **do**
   Select guide $g$ from $\mathcal{A}$; $v_i \leftarrow \omega v_i + c_1r_1(p_i - \alpha_i) + c_2r_2(g - \alpha_i)$; $\alpha_i \leftarrow \alpha_i + v_i$; **if** $\alpha_i$ *dominates $p_i$* **then**
    $p_i \leftarrow \alpha_i$;
   **end**
  **end**
  **if** *convergence reached* **then**
   **break**
  **end**
**end**
**return** $\mathcal{A}$ and images $\{\, \text{clip}(x + B\alpha, 0, 1) \,|\, \alpha \in \mathcal{A} \,\}$;

The resulting Pareto front reveals the feasible trade off surface between adversarial effectiveness and radiomic fidelity. In cases where no feasible adversarial exists under the FRD constraint, the front degenerates, indicating intrinsic robustness to radiomic preserving perturbations.

## 4. Experiments and Results

### 4.1. Experimental Setup

**Datasets.** We evaluate our approach on dermatology: datasets HAM10000 (HAM) (Tschandl et al., 2018), Dermofit (DMF) (Ballerini et al., 2013), and Derm7pt (D7P) (Kawahara et al., 2018). All images are preprocessed and normalized to the range $[0, 1]$ with a resolution of $224 \times 224$. Each image serves as a clean reference $x$, while perturbations are generated in the DCT space under the FRD constraint.

**Embedding Models.** We employ multiple pretrained embedding models to compute adversarial distances. Specifically, we utilize dermatology-focused models such as the *Google Derm Model* and *PanDerm* (Kiraly et al., 2024; Yan et al., 2024), alongside a general purpose vision encoder *CLIP* (Radford et al., 2021) pretrained on large-scale image datasets. Each model serves as an embedding backbone $f_\theta$, providing a stable feature space for measuring perturbation sensitivity. For a clean input $x$ and its adversarial variant $x'$, the adversarial embedding distance is defined as:

$$\mathcal{L}_{adv} = \|f_\theta(x) - f_\theta(x')\|_2. \tag{8}$$

**Multi-Task Heads.** Given an embedding $z$ from the backbone, we attach two lightweight heads: a classification MLP and a segmentation decoder. Both heads share the same embedding, allowing the pretrained encoder to retain rich features while adapting to task-specific outputs.

$$h_{\text{cls}}(z) = W_2, \sigma(W_1 z + b_1) + b_2, \tag{9}$$

$$M_{\text{seg}}(z) = \text{Decoder}(\text{Proj}(z)), \tag{10}$$

where $W_1, W_2, b_1, b_2$ are learnable parameters, $\sigma$ is ReLU, $\text{Proj}(\cdot)$ denotes a linear projection of $z$, and $\text{Decoder}(\cdot)$ is a lightweight transposed convolutional network producing a single-channel mask $M_{\text{seg}} \in \mathbb{R}^{B \times 1 \times 224 \times 224}$.

**Baseline Adversarial Vulnerability.** To establish a baseline of adversarial vulnerability in dermatological imaging, we evaluate the adversarial robustness of the above models using well known adversarial attacks, These attacks included pixel-domain methods FGSM, PGD, frequency-domain methods FGSM-DCT, PGD-DCT, and a black-box directional search method SIMBA (Goodfellow et al., 2015; Madry et al., 2017; Guo et al., 2019). All adversarial examples were generated targeting the OpenAI CLIP model, All perturbations were constrained to satisfy the $\ell_\infty$ norm bound, i.e., $\|\sigma\|_\infty \leq 0.05$. and then evaluated for zero-shot transferability across the other two models (Table 1). We assessed these attacks for both their ability to reduce model accuracy and the resulting FRD (Table 2), which quantifies to what extent radiomic fidelity is preserved or destroyed by the perturbations.

**MOPSO Attack.** We use the Multi Objective Particle Swarm Optimization (MOPSO) framework described in Equation 1 with a swarm size of 50, $d = 256$ DCT basis components, inertia weight $w = 0.7$, and coefficients $c_1 = c_2 = 1.5$. The number of iterations is set to 60. Perturbation strength is limited to $\epsilon = 0.05$ in pixel space.

## 4.2. Evaluation Metrics

We quantify results along two main axes:

- **Radiomic Fidelity (FRD)**, lower FRD values indicate closer alignment to the clean radiomic manifold.

- **Accuracy**, The proportion of correct predictions, including both true positives and true negatives.

The optimization aims to find the Pareto front that minimizes FRD while maximizing $\mathcal{L}_{adv}$, revealing whether adversarial examples can exist under radiomic constraints.

## 4.3. Results and Analysis

**Extreme Trade-off Between Attack Success and Radiomic Fidelity**
The results demonstrate a severe trade-off: attacks that successfully compromise model accuracy do so by violently disrupting the radiomic consistency of the images.

- **Violent Distribution Shift (FGSM/PGD):** The standard, full-space attacks (**FGSM** and **PGD**) achieved high attack success rates but at the cost of catastrophic radiomic integrity (Table 1). For example, the **CLIP** model on the DMF dataset suffered the lowest accuracy (19.34%, Table 1), coinciding with the highest FRD of $\mathbf{2.85 \times 10^8}$ (Table 2). This confirms that these highly successful attacks exist far outside the natural radiomic manifold.

- **DCT Constraint Reduces Attack Efficacy:** Attacks constrained to the low-frequency DCT subspace (**FGSM-DCT, PGD-DCT**) yielded FRD values orders of magnitude lower (Table 2) compared to their image-space counterparts. This preservation of radiomic fidelity came with a reduction in attack success. For instance, PGD-DCT on **CLIP/DMF** resulted in 31.69% accuracy (Table 1), significantly higher than the 19.34% achieved by the PGD attack, suggesting that enforcing radiomic preservation restricts the viable attack space.

Table 1: Accuracy under different adversarial attacks ($\downarrow$). Bold indicates the minimum accuracy per dataset column for each attack.

| Attack | Google Derm (%) | | | PanDerm (%) | | | CLIP (%) | | |
|---|---|---|---|---|---|---|---|---|---|
| | HAM | DMF | D7P | HAM | DMF | D7P | HAM | DMF | D7P |
| **Normal** | 90.03 | 74.49 | 77.20 | 91.12 | 81.48 | 74.09 | 89.76 | 80.66 | 73.06 |
| PGD-DCT | 82.68 | 76.54 | 72.02 | **78.06** | 79.84 | 73.06 | 83.77 | 31.69 | 67.36 |
| PGD | 80.42 | **48.15** | 66.84 | 79.15 | **32.92** | 63.73 | 67.27 | **19.34** | **55.96** |
| FGSM-DCT | 82.50 | 75.31 | 70.98 | 78.33 | 80.25 | 72.54 | 84.22 | 74.49 | 67.88 |
| FGSM | **68.18** | 49.38 | 63.21 | 80.60 | 53.56 | 63.21 | 78.88 | 48.15 | 60.62 |
| SIMBA | 77.75 | 55.97 | **61.66** | 78.15 | 56.38 | **62.18** | **64.73** | 27.57 | 59.07 |

Table 2: Fréchet Radiomic Distance (FRD) Scores for Classical Attacks.

| Dataset | FGSM (FRD) | PGD (FRD) | FGSM-DCT (FRD) | PGD-DCT (FRD) | SIMBA (FRD) |
|---|---|---|---|---|---|
| **HAM10000** | $1.55 \times 10^9$ | $3.81 \times 10^8$ | **38.07** | 146.23 | 174.70 |
| **DMF** | $7.29 \times 10^8$ | $2.85 \times 10^8$ | 14.21 | **12.51** | 159.77 |
| **Derm7pt** | 941.72 | 70483.44 | **24.94** | 19.36 | 112.14 |

## 4.4. Effect on Downstream Tasks

We assessed the impact of MOPSO-generated adversarial examples 3, constrained by FRD, on both classification and segmentation tasks across three datasets (HAM10000, DMF, Derm7pt) and three models (Google Derm, PanDerm, CLIP). As shown in Table 3. Despite these perturbations, classification performance is relatively maintained compared to results in Table 1. For instance, HAM10000 exhibits misclassification rates of 16.33% for Google Derm, 10.20% for PanDerm, and 12.24% for CLIP, whereas Derm7pt shows minimal impact, with rates of 1.75%, 0%, and 5.2%, respectively. DMF demonstrates a slightly higher sensitivity, particularly for Google Derm (21.74%), but overall, accuracy remains robust across all models.

Segmentation performance under MOPSO perturbations shows mostly small reductions in Dice score for HAM10000 and CLIP models (3.68–5.45%). PanDerm exhibits a larger drop (22.21%), indicating variability in the sensitivity of different datasets and models to FRD-constrained perturbations. Overall, these results suggest that while radiomic constraints limits adversarial effectiveness in some cases, certain models or datasets may still experience notable performance degradation.

Table 3: MOPSO drop in classification accuracy (%) vs FRD across datasets. FRD is reported once for single-image to batch and batch-to-batch comparisons, while misclassification rates are shown per model.

| Dataset | Single-image to batch FRD | Batch-to-Batch FRD | Misclassification (%) | | |
|---|---|---|---|---|---|
| | | | Google Derm | PanDerm | CLIP |
| HAM10000 | 5.994 | 0.694 | 16.33 | 10.20 | 12.24 |
| DMF | 5.815 | 0.908 | 21.74 | 8.70 | 14.49 |
| Derm7pt | 6.698 | 0.519 | 1.75 | 0 | 5.26 |

Table 4: MOPSO Segmentation drop in Dice score (%) vs FRD. FRD is reported once for single-image to batch and batch-to-batch comparisons, while Dice drop is shown per model.

| Dataset | Single-image to batch FRD | Batch-to-Batch FRD | Dice Drop (%) | | |
|---|---|---|---|---|---|
| | | | Google Derm | PanDerm | CLIP |
| HAM10000 | 5.994 | 0.694 | 5.45 | 22.21 | 3.68 |

## 4.5. Feasibility of Radiomic-Constrained Attacks (MOPSO)

To investigate the true feasibility boundary, we applied our **Multi-Objective Particle Swarm Optimization (MOPSO)** framework to random representative samples of the test sets. This gradient-free approach was designed to explore the **Pareto front** between **Fréchet Radiomic Distance (FRD)** and **Adversarial Deviation ($\mathcal{L}_{adv}$)**.

### 4.5.1. THE EXISTENCE BOUNDARY

The optimization results, visualized in the Pareto Fronts (Figure 1), clearly define an "existence boundary" dictated by radiomic consistency:

1. **High Fidelity Constraint ($FRD \leq 0.05$):** Under strict radiomic realism, the MOPSO optimizer largely failed, with negligible adversarial deviation ($\mathcal{L}_{adv} < 0.5$ for CLIP), showing that preserving authentic radiomic features inherently limits adversarial success.

2. **Feasibility Trade-off:** Meaningful adversarial effects emerged only when the image was allowed to diverge from radiomic realism (higher FRD), highlighting that successful attacks require violating intrinsic radiomic statistics.

### 4.5.2. OPTIMIZATION PERFORMANCE

Figure 2 shows that across datasets, the optimizer quickly reduces FRD during the early iterations, demonstrating its ability to generate radiomically consistent perturbations. Simultaneously, $\mathcal{L}_{adv}$ increases gradually, indicating the optimizer's trade-off between preserving radiomic fidelity and maximizing adversarial effect.

These results complement the Pareto fronts shown in Figure 1, which summarize the best achievable trade-offs between FRD and adversarial strength. The convergence plots highlight that, for low FRD targets, the optimizer often reaches a plateau with minimal adversarial gain, confirming the existence of a natural feasibility boundary. Conversely, allowing slightly higher FRD enables greater adversarial deviation, demonstrating the intrinsic trade-off between radiomic realism and attack effectiveness.

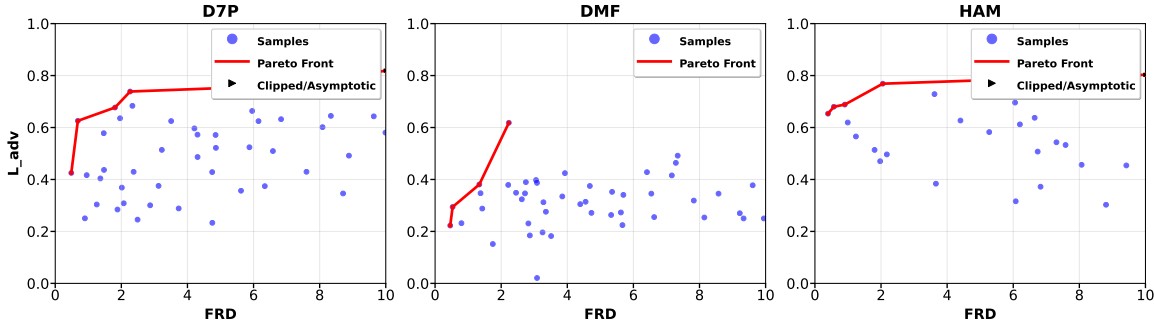

Figure 1: Pareto fronts of FRD vs. adversarial distance.

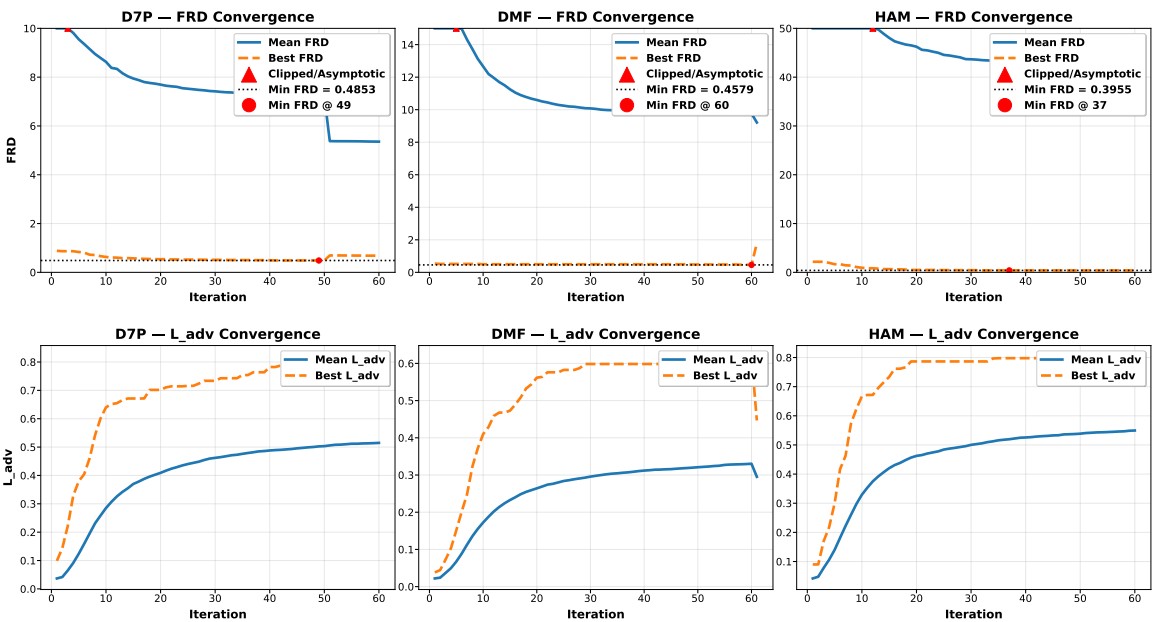

Figure 2: Convergence of minimal FRD and maximal adversarial distance during MOPSO optimization.

## 4.6. Discussion

The combined results of the classical attack baselines and the MOPSO feasibility study lead to a critical conclusion: standard adversarial attacks overestimate model vulnerability because they fail to account for radiomic plausibility. **Radiomic Consistency** acts as an intrinsic defense mechanism. The extremely high FRD scores in Table 2 for unconstrained attacks (FGSM/PGD) indicate that a successful attack trajectory often requires moving the image out of the radiomic manifold. Conversely, when we enforce this constraint through MOPSO, the "attack surface" shrinks dramatically, confirming that purely black-box adversarial examples that preserve clinical and radiomic fidelity are exceptionally difficult to realize.

## 5. Conclusion

This work presents the first systematic study of adversarial attacks constrained by radiomic fidelity, revealing that radiomic consistency imposes a natural boundary that renders meaningful adversarial attacks infeasible. Across multiple dermatological datasets and architectures, perturbations that preserve clinical plausibility (FRD $\leq$ 0.05) consistently fail to reduce the prediction accuracy, showing that the very properties that make medical images interpretable, textural, structural, and statistical coherence also confer intrinsic robustness. Our findings challenge the prevailing view of universal adversarial vulnerability. While unconstrained attacks succeed by producing images far outside the natural distribution,

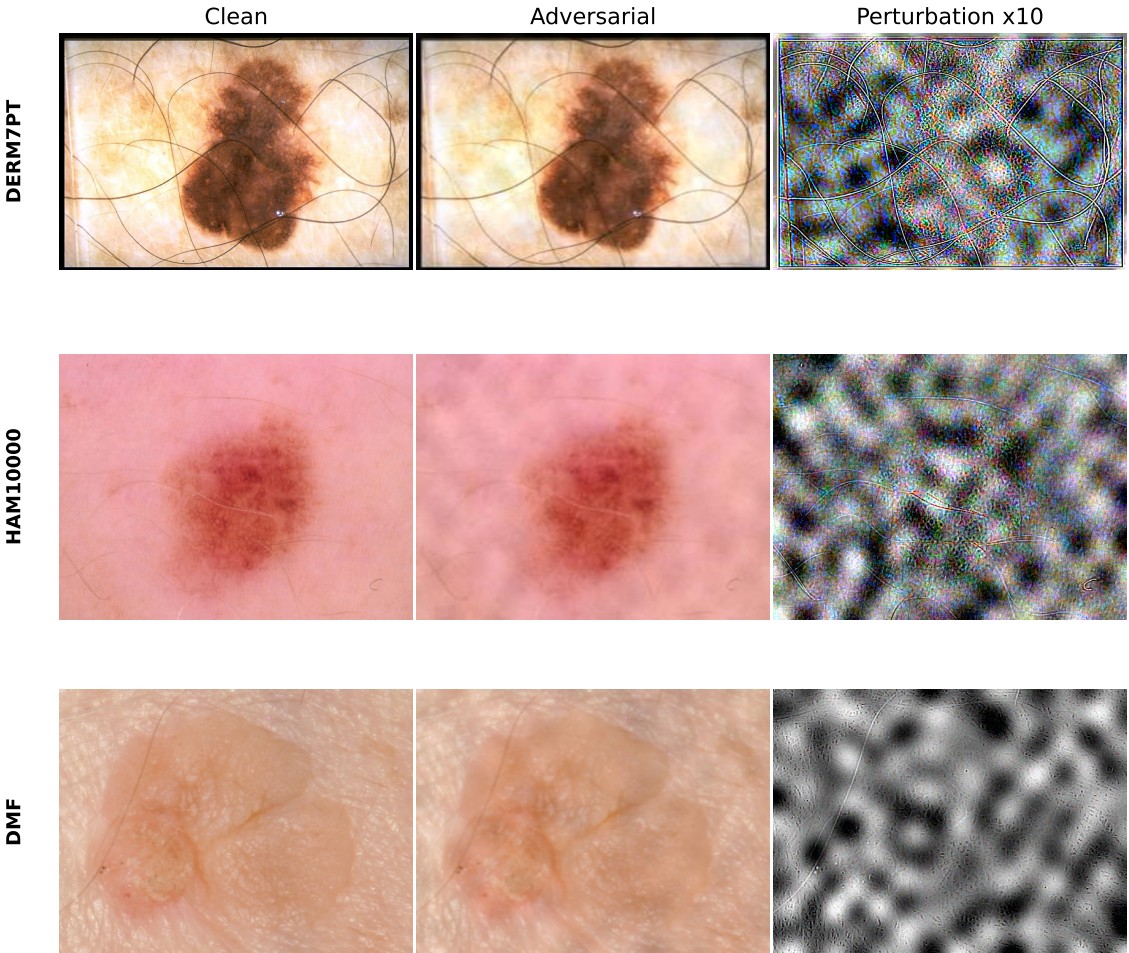

Figure 3: Comparison of clean images (left) and FRD-constrained adversarial images (center), with the corresponding perturbations visualized on the right, scaled by a factor of 10 for clarity.

radiomically constrained attacks collapse entirely, highlighting that real-world vulnerability in medical imaging is largely an artifact of unrealistic threat models. By formalizing adversarial feasibility within the radiomic manifold, this study opens a path toward trustworthy AI. Future work can extend radiomic constraints across modalities, derive provable robustness guarantees, and develop models that are robust, interpretable, and clinically aligned. Radiomic fidelity is not just desirable, it is shown to be a fundamental barrier to adversarial attacks. We hope our work motivates researchers to explore robustness grounded in clinical plausibility, design defenses that respect the natural manifold of medical images, and ultimately build AI systems that are trustworthy by design.

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
