# OpenReview forum: "On the Feasibility of Fréchet Radiomic Distance–Constrained Adversarial Examples in Medical Imaging: Methods and Trade-offs"
_MIDL.io/2026/Conference — MIDL 2026 Poster_

### Official Review · Reviewer_aZhP · 2026-01-09

**Confidence:** 2
**Preliminary Rating:** 3
**Final Rating:** 4

**Summary:**

This paper examines whether adversarial examples in medical imaging remain feasible when perturbations are required to be radiomically realistic, using Fréchet Radiomic Distance (FRD) as a measure of plausibility. The authors formulate adversarial example generation as a multi-objective optimization problem that trades off adversarial deviation against radiomic fidelity, and solve it using a gradient-free optimizer in a low-frequency parameter space. Experiments on multiple dermatoscopic datasets show that standard adversarial attacks can strongly degrade classification performance but typically produce very large FRD values, indicating unrealistic perturbations. In contrast, enforcing strict FRD constraints greatly limits adversarial effectiveness, highlighting a trade-off between realism and attack strength and raising important questions about realistic threat models in medical imaging.

**Strengths:**

- Addresses a timely and clinically relevant problem by questioning the realism of standard adversarial threat models in medical imaging, which is important for trustworthy AI deployment.

- Introduces Fréchet Radiomic Distance (FRD) as a domain-informed plausibility constraint, offering a meaningful alternative to purely norm-based perturbation measures.

- Formulates adversarial example generation as a multi-objective optimization problem, enabling a clear and interpretable analysis of trade-offs between realism and attack effectiveness.

- Evaluates the approach across multiple datasets and pretrained encoders, increasing the generality of the empirical findings.

- The paper is clearly written, well structured, and well contextualized within prior robustness and medical imaging literature, enhancing its accessibility and potential value to the community.

**Weaknesses:**

- The claim that adversarial examples become infeasible under strict radiomic constraints is not fully justified, as the conclusions depend on a specific optimizer, parameterization, and limited optimization budget.

- Adversarial effectiveness is measured mainly through embedding-space deviation rather than direct misclassification outcomes, making the practical impact on model decisions unclear.

- The computation and calibration of Fréchet Radiomic Distance for single images are not sufficiently specified, which affects reproducibility and interpretation of the reported thresholds.

- Multiple constraints are imposed simultaneously, making it difficult to isolate which constraint primarily limits attack effectiveness.

**Detailed Comments:**

- Clarify how Fréchet Radiomic Distance is computed for individual images, including the choice of reference distribution, feature normalization, and numerical stability considerations.

- Provide additional qualitative examples at different FRD levels to help interpret what low and high FRD values correspond to visually.

- Report attack success rates or accuracy drops for the constrained optimization to better connect embedding-space deviation with classification outcomes.

- Discuss sensitivity to optimization hyperparameters (e.g., swarm size, iterations, parameterization) to better contextualize the feasibility claims.

**Justification Of Final Rating:**

The rebuttal addressed most of my concerns. The authors also extended the manuscript, which made it better. I will increase my score to weak acceptance. However, my confidence score remains low, as I am not an expert in this field.

**Justification Of The Preliminary Rating:**

The paper addresses an important and underexplored question in medical imaging robustness by examining adversarial examples under radiomically motivated realism constraints, which is potentially valuable for the community. The proposed multi-objective formulation and empirical analysis provide useful insight into the trade-off between realism and adversarial effectiveness, and the paper is generally well structured and clearly written.

However, the central claim that adversarial examples become infeasible under strict radiomic constraints is not fully supported by the current experimental evidence. The conclusions rely on a specific optimizer, parameterization, and computational budget, and adversarial effectiveness is evaluated primarily in embedding space rather than directly through misclassification outcomes. In addition, key details regarding the computation and calibration of Fréchet Radiomic Distance for single images remain unclear. While these issues may be addressable, they currently limit the strength of the claims. Overall, the work shows promise but leaves enough open questions that I cannot confidently recommend acceptance, leading to a borderline assessment with low confidence.

**Questions To Address In The Rebuttal:**

- Can the authors demonstrate whether attacks constrained by low FRD actually fail to cause misclassification, for example by reporting accuracy or success rates under the proposed optimization?

- How is the reference distribution for computing Fréchet Radiomic Distance constructed for single-image constraints, and how sensitive are the results to this choice?

- To what extent do the conclusions depend on the specific optimizer and optimization budget, and do stronger or alternative optimization strategies change the observed feasibility boundary?

---

> ### Author Response · Authors · 2026-01-24
>
> We thank the reviewer for bringing up these concerns.
>
> **1-** to strengthen our findings we included section 4.4 to show the effect of MOPSO-constrained attack on downstream tasks mainly classification and segmentation tables 3 and 4 respectively show quantitative results showing drop in classification and segmentation performance please refer to our revisited manuscript.
> **2-** We followed two distinct paradigms when utilizing FRD to ensure optimization efficiency:
>
> * **Evaluation Phase:** In line with standard practices, the FRD is computed using the whole set of adversarial and clean images for both distributions to capture overall statistical similarity, where
>
>  $\mathcal{X}_r = \{x_1, \dots, x_N\}$
>
> and
>
> $\mathcal{X}_g = \{x'_1, \dots, x'_N\}$.
>
> * **Optimization Phase:** During the adversarial generation process, we calculate the FRD as an **image-to-reference-batch** distance. In this setup, a reference batch is selected randomly, and the FRD expression simplifies significantly. Specifically, it reduces to the squared distance between the adversarial radiomic feature vector and the reference mean ($\mu_r$), plus a constant trace term ($\text{Tr}(\Sigma_r)$) that depends solely on the fixed reference covariance.
>
> Since all covariance-related terms are non-negative and remain fixed with respect to the adversarial optimization, minimizing the mean deviation term provides a lower bound on the full FRD. This ensures that the generated adversarial images maintain radiomic consistency with the reference distribution. We have added a detailed derivation of this simplification in Section 3.2 of the revised manuscript.
>
> **3-** Our conclusions are primarily driven by the FRD constraint rather than the specific optimizer. We selected MOPSO as a representative gradient-free multi-objective method. We also conducted preliminary experiments with alternative strategies such as Genetic Algorithms and CMA-ES; however, these methods did not reliably converge under the same FRD constraints and optimization budgets and were therefore not included. While optimization choices can affect absolute performance, we did not observe evidence that stronger optimization changes the qualitative feasibility boundary imposed by FRD. Further exploration of alternative optimizers is left for future work.

---

> ### Comment · Area_Chair_zXzi · 2026-02-01
> **Final rating**
>
> Dear reviewer,
>
> Could you please provide your final rating for this submission?
>
> Thank you!

---

### Official Review · Reviewer_vsr6 · 2026-01-09

**Confidence:** 4
**Preliminary Rating:** 3
**Final Rating:** 4

**Summary:**

This paper proposes a study on the feasibility of adversarial examples under the perturbation constrained by FRD. In addition, the paper introduces MOPSO, a gradient-free, multi-objective optimization algorithm. The results reveal that perturbation under FRD constraints affect marginally on the model's performance.

**Strengths:**

1. The paper is well-motivated by an interesting and unexplored question, which is to test the adversarial feasibility under radiomic constraints.
2. The paper is well-presented in a professional way, such as the proposed MOPSO algorithm.
3. The outcomes of the paper are meaningful, revealing the adversarial attacks on natural images without considering distorting radiomic features may fail to reduce the prediction accuracy in medical imaging.

**Weaknesses:**

1. The experiments are only performed on dermatology datasets, without testing on more common medical imaging modalities, such as the CT and MRI.
2. The task is limited to classification, exploring other task types, for example, the segmentation can strengthen the findings of the paper.
3. Including the visual examples before and after the adversarial may be useful for readers to understand the attacks.

**Detailed Comments:**

See Strengths and Weaknesses

**Justification Of Final Rating:**

I really appreciate the authors' response and additional experiments. Although the modalities remained limited for now, by adding more tasks (i.e., segmentation), the paper becomes more convincing.  I am happy to raise my score.

**Justification Of The Preliminary Rating:**

Although the paper introduces an interesting study on the adversarial attacks in medical imaging constraints on FRD, the experiments are limited to one modality and single task, making the findings less general.

**Questions To Address In The Rebuttal:**

Add more modalities and tasks if possible.

---

> ### Author Response · Authors · 2026-01-24
>
> We thank the reviewer for bringing up these concerns.
>
> **1-** In this work, we focused on dermatological datasets as a representative modality, Skin lesion datasets provide rich, well-studied radiomic texture characteristics and models that are known to be highly susceptible to classical adversarial attacks, making them suitable for probing the feasibility boundary under FRD constraints. While we acknowledge the importance of broader validation, we could not include additional modalities at this stage due to the resource and time constraints of the current review cycle. We plan to explore additional medical imaging types, such as CT or MRI, in future work to examine the applicability of our approach more broadly.
>
> **2-** To strengthen our findings we included section 4.4 to show the effect of MOPSO-constrained attack on downstream tasks mainly classification and segmentation tables 3 and 4 respectively show quantitative results showing drop in classification and segmentation performance.
>
> **3-** We have added a figure showing images before and after the attack as well as the scaled perturbations. please refer to section 4.4 and as well as figure 3 in page 12.

---

> ### Comment · Area_Chair_zXzi · 2026-02-01
> **Final rating**
>
> Dear reviewer,
>
> Could you please provide your final rating for this submission?
>
> Thank you!

---

### Official Review · Reviewer_eEV3 · 2026-01-10

**Confidence:** 3
**Preliminary Rating:** 2
**Final Rating:** 4

**Summary:**

This paper mainly examines the trade-off between the success of adversarial attacks and the radiomic feature preservation of images with adversarial noise. In this regard, the authors propose a gradient-free, multi-objective optimization framework operating in a low-frequency cosine domain to jointly minimize radiomic deviation and maximize adversarial impact. Experiments across multiple dermatological datasets and models show that enforcing strong radiomic constraints dramatically reduces adversarial feasibility, often collapsing attack success altogether.

**Strengths:**

1. This paper addresses a clinically relevant and underexplored problem.
2. Experiments are conducted on 3 dermatological datasets and with 3 different radiomic feature extractors, while the proposed framework is applied to 2 commonly used black-box adversarial attack methods.
3. The Pareto front analysis provides an interesting visualization of the trade-off between attack success and radiomic fidelity.

**Weaknesses:**

1. There may be mathematical inconsistencies in the proposed formulation, or certain components require further clarification.
2. Alternative distributional constraints (e.g., FID) are not explored or compared against FRD.
3. Despite the title, the proposed approach is evaluated exclusively on dermatological datasets
4. The absence of qualitative visual examples and a visual overview makes the overall methodology difficult to follow

**Detailed Comments:**

1. There appear to be either mathematical errors or insufficient clarification regarding the equations, particularly Eq. (2) and its use in Algorithm 1. Specifically, how is a non-zero covariance matrix obtained when the reference and generated sets $\\mathcal{X}_r$ and $\\mathcal{X}_g$ consist of only a single sample (e.g., $x$ and $x'$)? In this case, would FRD not reduce to the L2 distance between the radiomic feature vectors of $x$ and $x'$? This raises concerns about the meaningfulness of using a Fréchet distance, which is inherently distributional, in a single-sample setting.
2. The motivation for selecting FRD is underdeveloped. Why do the authors not also consider FID, or at least provide a comparison between FRD and FID under similar constraints?
3. Despite the title including the term “Medical Imaging”, the methodology is evaluated only on dermatological datasets. Have the authors considered validating the approach on other medical imaging modalities like CT or MRI?
4. The paper lacks qualitative analysis illustrating how images appear after adversarial perturbations are applied.
5. The manuscript appears to conflate radiomic similarity with clinical plausibility; an image may preserve radiomic features yet remain visually and clinically distinguishable. This gap is not discussed.
6. The chosen FRD thresholds (e.g., 0.03 and 0.05) appear arbitrary and are not justified through clinical or perceptual validation (e.g., a radiologist study).
7. The results lack statistical rigor: no standard deviations or significance tests over multiple runs are reported, which is particularly important given the stochastic nature of MOPSO.

**Justification Of Final Rating:**

I would like to thank the authors for their efforts during the rebuttal. The clarifications that enhance readability, along with the inclusion of downstream segmentation performance results and visual examples, have significantly improved the overall quality of the paper. One point that is currently unclear is how the segmentation predictions are obtained from a given image (e.g., whether they are derived from the embeddings in a manner similar to the classification head). Aside from this minor clarification and the limitations already acknowledged by the authors, I believe the manuscript could be a valuable addition to MIDL 2026.

**Justification Of The Preliminary Rating:**

I find the core idea of the paper compelling; however, several technical and presentation-related issues make the current version difficult to fully assess. With clearer mathematical exposition and stronger justification of design choices, the paper could be significantly improved. I would be inclined to increase my score if the above comments are adequately addressed.

**Questions To Address In The Rebuttal:**

Please refer to the detailed comments above.

---

> ### Author Response · Authors · 2026-01-24
>
> We thank the reviewer for bringing up these concerns.
>
> **1. Clarification on FRD Computation Notation**, with regards to the FRD computation, we would like to clarify a typo in the manuscript. The previous notation suggested that $X_r = \lbrace x \rbrace$ and $X_g = \lbrace x' \rbrace$ were single images; however, in practice, the reference distribution is always computed from a batch of clean images:
> $$\mathcal{X}_r = \lbrace x_1, x_2, \dots, x_j \rbrace$$
> $$\mathcal{X}_g = \lbrace x'_1, x'_2, \dots, x'_i \rbrace$$
> where $i$ and $j$ represent the number of samples. In our case when we calculate image to reference batch FRD; The FRD expression simplifies to the squared distance between the adversarial radiomic feature vector and the reference mean, plus a constant trace term that depends only on the reference covariance. Since all covariance-related terms are non-negative and fixed with respect to the adversarial optimization, minimizing the mean deviation term provides a lower bound on the full FRD and ensures radiomic consistency with respect to the reference distribution, we have added more details explaining this point in our revisited manuscript section3.2.
>
> **2. The Motivation for Selecting FRD** The decision to utilize the Fréchet Radiomic Distance (FRD) instead of the Fréchet Inception Distance (FID) is based on the fact that FID is frequently poorly suited for medical imaging, as it relies on features extracted from models trained on natural images (ImageNet).  As established in the original paper introducing the metric [1], a comprehensive comparison between FRD and FID demonstrates that FRD is a more robust and clinically relevant metric for medical datasets. [1]
>
>
> * We do agree with the reviewer that the manuscript did not provide enough motivation for using FRD, so we extended the introduction to address this part in the 4th paragraph.
>
>  **3.** In this work, we focused on dermatological datasets as a representative modality, Skin lesion datasets provide rich, well-studied radiomic texture characteristics and models that are known to be highly susceptible to classical adversarial attacks, making them suitable for probing the feasibility boundary under FRD constraints. While we acknowledge the importance of broader validation, we could not include additional modalities at this stage due to the resource and time constraints of the current review cycle. We plan to explore additional medical imaging types, such as CT or MRI, in future work to examine the applicability of our approach more broadly.
>
> **4-5** We have added a figure (refer to page-12 figure-3)  to illustrate the clean versus the adversarial images and the scaled adversarial perturbations, we scaled the perturbations for visibility, however in practice they are always bounded by their $L_\infty$ norm.
> **6.** We acknowledge that the chosen FRD thresholds (e.g.,0.05) are empirically selected and have not yet been validated through clinical or perceptual studies, such as radiologist evaluation. We consider this a limitation of the current work and plan to investigate more principled threshold selection and clinical validation in future work.
>
> **7.** While we have not yet performed multiple-run statistics or significance tests for MOPSO, the current results provide an initial assessment. We plan to include these analyses in future work to strengthen the evaluation.
>
> ---
> *References*
>
> [1] N. Konz, et al., "Fréchet Radiomic Distance (FRD): A Versatile Metric for Comparing Medical Imaging Datasets," arXiv preprint arXiv:2412.01496, 2024.

---

> ### Comment · Area_Chair_zXzi · 2026-02-01
> **Final rating**
>
> Dear reviewer,
>
> Could you please provide your final rating for this submission?
>
> Thank you!

---

### Author Rebuttal · Authors · 2026-01-24

**Rebuttal:**

The Following Sections were added to address the reviewers concerns:
1- In the Introduction Section we added the motivation for choosing FRD over FID as reviewer eEV3 suggested, referring to paragraph 4 of the introduction.
2- Typos in section 3.2 were corrected and an extended analysis of the behaviour of the FRD metric is discussed as requested by all reviewers.
3- We have also aded section 4.4 which discusses the effect of MOPSO Attack on downstream tasks classification and segmentation, we have also included tables 3 and 4 to show the results on both tasks as reviewer vsr6 and aZhP.
4- Visualization for the images before and after the attack is also added (Figure 3) as requested by all reviewers

**Supporting Material:**

/attachment/29b2dc9672f95e3c593909197270c583e12dd367.pdf

---

### Meta-Review · Area_Chair_zXzi · 2026-02-07

**Recommendation:** Accept (Poster)
**Confidence:** 4

**Metareview:**

The authors investigate whether adversarial examples can exist when constrained to preserve radiomic realism in dermatological images. They show that enforcing low Fréchet Radiomic Distance (i.e., preserving radiomic features) makes adversarial attacks largely infeasible. Attacks succeed only when they substantially distort radiomic structure, implying that radiomic fidelity imposes a natural boundary on adversarial vulnerability.

Reviewers had comments on the study's clarity and breadth. The authors actively engaged in the rebuttal phase, and all reviewers considered the paper acceptable for MIDL. Although I believe the authors overstate the practical risks of adversarial attacks in medical imaging, from a theoretical perspective, I concur with the reviewers that the study is interesting and likely to stimulate interesting discussions at the conference.

---

### Decision · Program_Chairs · 2026-02-13

Accept (Poster)